# Terrestrial ion escape and relevant circulation in space

Masatoshi Yamauchi

Swedish Institute of Space Physics, Box 812, S-98128 Kiruna, Sweden **Correspondence:** M. Yamauchi (M.Yamauchi@irf.se)

**Abstract.** Observations of the terrestrial ion escape to space the transport of escaping ions in the magnetosphere are reviewed, with main stress on subjects that were covered in reviews over past two decades although Cluster significantly improved out knowledge of them. Here, outflowing ions from the ionosphere are classified in terms of energy rather than location: (1) as cold ions refilling the plasmasphere faster than Jeans escape, (2) as cold supersonic ions such as the polar wind, and (3) as

suprathermal ions energized by wave-particle interaction or parallel potential acceleration, mainly starting from cold supersonic ions. The majority of the suprathermal ions above the ionosphere become "hot" at high altitudes with much higher velocity than the escape velocity even for heavy ions. This makes heavy hot ions more abundant in the magnetosphere than heavy ions transported by cold refilling ions or cold supersonic flow.

The immediate destination of these terrestrial ions varies from the plasmasphere, the inner magnetosphere including those entering the ionosphere in the other hemisphere and the tailward outer boundaries, the magnetotail, and the solar wind (magnetosheath and cusp/plasma mantle). Due to time variable return from the magnetotail, ions with different routes and energy meet in the inner magnetosphere, making it a zoo of different types of ions in both energy and energy distribution. While the massindependent drift theory has successfully disentangled this zoo of ions, there are many poorly understood phenomena, e.g., mass-dependent energization. Half of the heavy ions in this zoo also finally escape to space, mainly due to magnetopause shad-

owing (overshooting of ion drift beyond the magnetopause) and charge exchange near the mirror altitude where the exospheric neutral density is at its highest.

The amount of heavy ions mixing directly with the solar wind is already the same or larger than that entering into the magnetotail, and is large enough to extract the solar wind kinetic energy in the cusp/plasma mantle through the mass-loading effect and drive the current system near the cusp independently from the global current system. Considering the past solar and solar wind conditions, ion escape might even have influenced the evolution of the terrestrial biosphere.


Copyright statement. TEXT

### 1 Introduction

Circulation and roles of ionospheric heavy ions have long been an important subject in the magnetospheric physics since they are found almost everywhere in the magnetosphere (Chappell et al., 1972, Shelley et al., 1972), including the high-latitude

- magnetosheath and plasma mantle (Lundin, 1985, Eklund et al., 1997). There are many studies on this problem with many reviews, for examples, on refilling and dynamics of the plasmaspheric cold ions (Darrouzet et al., 2009; Welling et al., 2015), the polar wind and cold supersonic ion outflow (Yau et al., 2007; Moore and Horwitz, 2007; André, 2015), the other outflow from the ionosphere to the magnetosphere (Moore et al., 1999a; Lotko, 2007; Maggiolo, 2015; Welling et al., 2015), and plasma sheet dynamics including energization in the magnetotail and resultant ring current ions in the inner magnetosphere
- (Blanc et al., 1999; Walker et al., 1999; Ebihara and Ejiri, 2003; Kronberg et al., 2014). However, very few reviews mention the total escape rate of the ionospheric ions to space (Moore and Horwitz, 2007; André, 2015) and no review discusses its quantitative importance on the atmospheric evolution, in contrast to the cases for Mars (Jakosky et al., 2015). To estimate this "ion budget" problem, two key subjects have been missing from the reviews during the past two decades: the outflowing ions directly accessing the solar wind in the magnetosheath, cusp, and outer part of plasma mantle, and the fate of trapped hot ions
- that have much lower energy than the ordinary ring current, although a substantial amount of these ions will be lost to space (Dandouras et al., 2018).

The Cluster orbit and instrumentation allowed us to investigate these missing subjects with statistical significance, making it possible to synthesize these observations in the context of this ion budget problem. This paper synthesizes these missing parts that have been revealed by Cluster, to obtain the ion escape rate to space and its consequences that have been overlooked in

the past. Acceleration mechanisms for the outflowing ions such as wave-particle interactions (Lundin and Guglielmi, 2006), electrostatic field and centrifugal acceleration (Cladis, 1986) are not covered in this paper, and hence the roles of electrons (e.g., Strangeway et al., 2005) are not discussed. Similarly, numerical simulations of O<sup>+</sup> tracing (e.g., Moore et al., 2014) are not covered

Since the Cluster Ion Spectrometory COmposition DIstribution Function (CIS/CODIF) instrument is not designed to separate more than four main species H<sup>+</sup>, He<sup>++</sup>, He<sup>+</sup>, and atomic ions of the CNO group (Rème et al., 2001), all heavy ions are pedagogically called oxygen ions O<sup>+</sup> as is the usual convention, although nitrogen ions (N<sup>+</sup> and N<sub>2</sub><sup>+</sup>) become significant during geomagnetic storms (Hamilton et al., 1988; Yau and Whalen, 1992). The paper is organized based on my Bartels Medal lecture at EGU General Assembly 2019, as following with stress on Sects. 4 and 5.

- 1. Introduction
- 2. Ion outflow from the ionosphere
  - 3. Destinations of the outflow
  - 4. Inner magnetosphere at L < 6: zoo of many processes
  - 5. Consequences of large amount of direct ion escape
  - 6. Discussion
- 7. Conclusion

### 2 Ion outflow from the ionosphere

Outflow of ionospheric heavy ions are commonly observed at high latitudes (e.g., Moore et al., 1999a, André, 2015). These outflowing ions are often classified by the source localtion, but they can also be classified by detection method: (1) cold filling to the plasmasphere (Park, 1974; Welling et al., 2015), (2) cold supersonic outflow to the inner magnetosphere and magnetotail such as polar wind (Su et al., 1998; Engwall et al., 2006; 2009), and (3) ions with suprathermal energy above the

ionosphere (e.g., Eliasson et al., 1994) or with higher energy at higher altitude (e.g., Möbius et al., 1998). This paper follows this classification.

# 2.1 Cold filling


The cold filling flow along the magnetic field to the plasmasphere has long been implied from the time profile of the local

- plasmaspheric density with a drastic decrease during substorms and gradual recovery over days during quiet periods (Craven et al., 1997; Darrouzet et al., 2009; Sandel, 2011). In addition, continuous refilling of the plasmaspheric outward wind is also expected (Dandouras et al., 2013). Direct detection of refilling flow has been tried (Singh and Horwitz, 1992; Watanabe et al., 1992), using the distribution function method and an assumption of the same E×B drift velocity between different species to overcome the spacecraft potential. The estimated field-aligned flow velocities was less than 1 km/s (which means less than 0.02)
- 70 eV for He<sup>+</sup>), i.e., subsonic, while this could be underestimated (S. Watanabe, private communication, 2019) and this problem is still an open issue.

The refill rate has been estimated from the loss rate of the plasmaspheric ions for both sporadic plumes of  $10^{27}$  s<sup>-1</sup> for about 5-10% of time (Sandel and Denton, 2007; Darrouzet at al., 2008) and continuous loss of maximum  $5 \times 10^{26}$  s<sup>-1</sup> (Dandouras et al., 2013), leading to a total rate of about  $6 \times 10^{26}$  s<sup>-1</sup>. The composition is also estimated as 90% of thermal H<sup>+</sup> and 10% of

He<sup>+</sup>, with only 1-5% of O<sup>+</sup> abundance (Darrouzet et al., 2009; Gallagher et al., 2016). The abundance of O<sup>+</sup> in the refilling flux indicates that the process is much more effective than that of the Jeans escape. However, the refilling mechanism is not yet completely understood (Darrouzet et al., 2009; Gallagher et al., 2016).

## 2.2 Cold supersonic outflow

The cold supersonic outflow here means the flow of ions with kinetic energy much higher than thermal energy but lower than the satellite potential when it is positive, i.e., inside dense plasma under sunlit conditions. For the Cluster case, active potential control did not help detect it (Sauvaud et al., 2004). In addition to the methods that are described in the previous subsection, Engwall et al. (2006) found a new method to obtain the bulk velocity and flux of this outflowing cold supersonic outflow in the lobe region where the density is very low. The obtained typical velocity is about 25 km/s (3 eV) at  $10-15 R_E$  from the Earth, as shown in Fig. 1 (Engwall et al., 2009). The increasing velocity with distance is consistent with centrifugal acceleration (Cladis,

1986), and with 10 km/s velocity at > 5000 km altitude in the dayside polar cap for  $H^+$  (Abe et al., 1993; Su et al., 1998) and hence the polar wind (Pollock et al., 1990; Yau et al., 2007, and references therein). This flow increases with increases of F10.7

Figure 1. Altitude dependence of outward parallel velocity of cold supersonic flow observed by Cluster (Engwall et al., 2009)

flux, solar wind dynamic pressure, Kp, and southward IMF ( $B_z$ ), in a consistent manner with DE-1 and Akebono observations (Yau et al., 2007 and references therein).

The O<sup>+</sup>/H<sup>+</sup> density ratio decreases quickly with altitude, while the upward velocity ratio  $(V_{O\parallel}/V_{H\parallel})$  gradually increases 90 toward the unity with altitudes above > 5000 km (Abe et al., 1993, Yau et al., 2007). Slow O<sup>+</sup> stays at low-altitude longer and thus experiences the wave-particle interaction longer than fast H<sup>+</sup>, and low-frequency waves accelerate different species by the same velocity kick (Lundin and Guglielmi, 2006). Therefore, the velocity-filtered O<sup>+</sup> has a velocity similar to H<sup>+</sup> at higher altitude, and a higher fraction of O<sup>+</sup> than H<sup>+</sup> becomes hot before it is convected to the lobe region.

### 2.3 Suprathermal and hot outflows

- The last category (suprathermal ions and hot ions) is the one that can be directly detected by the ion instruments. In this paper, suprathermal ions and hot ions are not subdivided because the former is further accelerated to become the latter at > 6 R<sub>E</sub> distance (Lennartsson et al., 2004; Arvelius et al., 2005; Nilsson et al., 2006) or during travel to the other hemisphere (Cattell et al., 2002; Hultqvist, 2002; Yamauchi et al., 2005b). On the other hand, cold supersonic outflow and hot outflow are well separated in the magnetotail or plasma sheet according to the very few direct observations that identified both components 100 (Olsen, 1992; Seki et al., 2003). The largest ionospheric sources of the suprathermal and hot ion outflows are around the
- dayside cusp and the nightside auroral oval (Moore et al., 1999a; Peterson et al., 2001).

For the dayside hot outflow, statistics at mid and high altitudes indicate that different species (H<sup>+</sup> and O<sup>+</sup>) are accelerated by the same velocity kick rather than by the same energy gain (Abe et al., 1993; Lennartsson et al., 2004; Nilsson et al., 2006). Polar apogee observations at 8 R<sub>E</sub>, which is normally in the polar cap, showed  $V_{O\parallel}/V_{H\parallel}$  about 0.3-0.6 (Su et al., 1998),

which is also larger than 0.25 that corresponds to the same energy. The outflow flux is higher for  $O^+$  than He<sup>+</sup> (e.g., Abe et al., 1993), with a flux ratio ( $F_O/F_H$ ) of about 0.1 for the suprathermal energy range (thermal ion instrument for < 50 eV) and is close to 1 for the hot ion energy range of > few eV up to tens of keV (Moore et al., 1999a; Curry et al., 2003; Peterson et al., 2001; Sandhu et al., 2016). Unless the velocity filter effect makes such a variation, all these observations indicate non-thermal

Figure 2. Energy-time spectrograms for heavy ions (marked as  $O^+$ ), protons, and electrons on 21 February 1994 observed by Freja in the dayside. Kp was 7+ and the IMF  $B_Y < -50$  nT, which moved the northern cusp dawnward and southern cusp duskward (Yamauchi et al., 2005b)

energization mechanisms (Moore et al., 1999a; Lennartsson et al., 2004; Lundin and Guglielmi, 2006; Waara et al., 2011) such as wave-particle interaction (with minor contribution by the centrifugal acceleration).

There is also one case study that observed the outflowing ions both before and after such accelerations, when the interplanetary magnetic field (IMF) was extremely dawnward. This condition shifted the northern cusp toward prenoon and the southern cusp toward postnoon, and hence the ion outflow from the southern cusp was detected as the ion inflow in the northern postnoon, well separated from the northern cusp signature, as shown in Fig. 2 (Yamauchi et al., 2005b).

- While the outflowing energy from the northern cusp is similar between  $O^+$  and  $H^+$ , the energy ratio of inflowing  $O^+$  and  $H^+$ 115 is nearly 15 when comparing the same location, and about  $\sim 20$  when comparing the energies of the most intense injections. Thus the ions are accelerated to the same velocity rather than the same energy in the magnetosphere even after considering the possible velocity filter effect. Therefore, the energization of the dayside hot ion outflow at mid- and high-altitudes must be mainly by waves or other non-thermal processes. This applies even at low-altitudes because the majority of the dayside 120 outflowing ions at low altitudes are conic-like at low altitude rather than beam-like (Norqvist et al., 1996; Peterson et al.,

2008).

On the other hand, a substantial portion of the nightside hot outflow from the auroral zone, for which the total amount is already one orders less than the dayside outflow from around the cusp and prenoon auroral region (Peterson et al., 2001; Yau et al., 2007 and references therein), is in the beam form (Norqvist et al., 1998; Peterson et al., 2006), and is easier to return back to the ionosphere in the other hemisphere. For the velocity ratio, we expect  $V_{O\parallel}/V_{H\parallel} = 0.25$  after acceleration by the parallel

electric potential, and hence O<sup>+</sup> moving more downstream than H<sup>+</sup> compared to the mapping location along the geomagnetic field due to the velocity filter effect.

#### 3 **Destinations of the outflow**

This section describes the immediate destinations of the outflow but not the final destinations. Unlike above the ionosphere, cold ions in the magnetosphere have been observed by traditional hot ion instruments and hence on the very few occasions 130

when the number density is high; for examples in the eclipse (Seki et al., 2003) and during bulk motion by Pc5 pulsation (Hirahara et al., 2004) or strong  $E \times B$  drift (Sauvaud et al., 2001, Yamauchi et al., 2009a).

### 3.1 Cold filling to plasmasphere

The destination of cold filling is mainly the plasmasphere as mentioned in Sect. 2.1. The refilling rate is estimated from 135 the recovery of plasmasphere after losing a massive amount of cold ions as plasmaspheric plumes (Sandel and Denton, 2007; Darrouzet at al., 2008) or outward wind (Dandouras et al., 2013) because they are most likely lost to space rather than returning.

#### 3.2 Known destination for cold supersonic outflow

The destination of the cold supersonic outflow is not yet clear except the tail plasma sheet because measurement is possible only in the low-density lobe region (Engwall et al., 2006; André, 2015). Engwall et al. (2009) derived the total flux flowing in the lobe and its Kp dependency and solar wind dependency. Their results can be scaled to roughly  $3 \exp(0.23 \text{Kp}) \times 10^{25} \text{ s}^{-1}$ 

for H<sup>+</sup>, with very low ( $\leq 1\%$ ) O<sup>+</sup>/H<sup>+</sup> ratio.

In the lobe, the plasma convection across the geomagnetic field cannot be ignored compared to the outflow velocity, making the destination significantly different between species by the velocity filter effect (Chappell et al., 1987) unless they accelerated to the same velocity. Therefore,  $O^+$  is bent more toward the lower latitude than  $H^+$  if they could reach the tail plasma sheet.

This explains the recent observations of higher  $O^+/H^+$  flux ratios of trapped ions at lower energy and at lower geocentric distance in the inner magnetosphere (Claudepierre et al., 2016; Kistler and Mouikis, 2016) despite the low  $O^+$  content in this flow, because adiabatic energization in the tail is larger for more distant start point, i.e., larger for  $H^+$  than  $O^+$  if the destination are different (Ejiri, 1978, Ebihara and Ejiri, 2003).

### 3.3 Various destinations for hot outflow

For the suprathermal and hot ions, the destination depends on the starting location and conditions in the ionosphere. Therefore, we consider the nightside outflow differently from the dayside outflow.

The dayside outflow has wide destinations, covering the tail plasma sheet, the plasma mantle, and even the magnetosheath (Shelley et al., 1972; Lundin 1985; Eklund et al., 1997). Among these destinations, direct loss to space through the plasma mantle and magnetosheath has been underestimated before the Cluster observations, which showed that the amount is signifi-

cant. The total hot O<sup>+</sup> flux into plasma mantle and magnetosheath is as large as  $10^{25-26}$  s<sup>-1</sup> (Nilsson et al., 2012; Slapak et al., 2017a) and is larger than the total hot O<sup>+</sup> flux into the tail plasma sheet (Slapak et al., 2017b, Slapak and Nilsson, 2018). Geotail found mantle-like populations in the distant-tail plasma sheet beyond -150 R<sub>E</sub> (Maezawa and Hori, 1998). They are flowing anti-sunward, suggesting that the heavy ions in the plasma mantle are generally lost to space.


The Cluster statistics of the solar (F10.7 flux), solar wind, and Kp dependences of these  $O^+$  outflows in these regions show that solar wind dynamic pressure (Fig. 3), solar wind coupling function, and Kp are the most influencing parameters, with 1.5 orders of magnitude difference between quiet and active cases (Slapak et al., 2017a; Schillings et al., 2019). The IMF Bz

**Figure 3.** Cluster statistics of hot heavy ions  $(O^+)$  escape to space in the plasma mantle and magnetosheath. Total escape flux of the heavy ions is plotted as a function of the solar wind dynamic pressure for different IMF clock angles (CA), which is defined as  $0^\circ$  for northward IMF (Schillings et al, 2019)

dependence is not as drastic as the dependence solar wind dynamic pressure, as shown in Fig. 3. These results are generally consistent with previous statistics of suprathermal or hot ion outflow at lower altitudes near the cusp (Moore et al., 1999a; Cully et al., 2003; Lennartsson et al., 2004).

- However, Schillings et al. (2019) also found that the dependence on F10.7 flux is very weak. This is quite different from drastic F10.7 dependence of the suprathermal O<sup>+</sup> outflow at lower altitudes (Cully et al., 2003) or cold supersonic flow (Engwall et al., 2009). The difference indicates the following scenario: the extra acceleration by the wave-particle interaction and the wave activity reaching there does not depend on solar EUV but strongly on the solar wind's mechanical (dynamic pressure) and electric (coupling function) energies. Ions that received such acceleration may reach the exterior cusp and the
- plasma mantle before the velocity filter effect bends them toward the nightside or low-latitudes. In fact, the X flares enhance only the ion density and temperature without enhancement of ion outflow, which increases after the arrival of solar energetic particle events or coronal mass ejections (Yamauchi et al., 2018).

For the  $O^+$  outflow from the nightside ionosphere, the majority of ion beams after being accelerated by the parallel electric potential drops are expected to re-enter the ionosphere in the opposite hemisphere as mentioned in Sect. 2.3. In fact, Freja at

175 1700 km altitude detected injecting ions with similar energy between H<sup>+</sup> and O<sup>+</sup> ( $V_{O\parallel}/V_{H\parallel} \sim 4$ ) in keV range, suggesting that they are accelerated by parallel electric potential (Hultqvist, 2002). The observation also indicates the velocity filter effect: the O<sup>+</sup> injection events are found mostly inside or equatorward of the H<sup>+</sup> injection events, ending up in what is traditionally