# Peer review of "Terrestrial ion escape and relevant circulation in space"

_Annales Geophysicae, 2019_

## Referee Comment (RC1) · Anonymous Referee #1 · 30 Jul 2019

This manuscript is reported to be based on the Bartels Medal Lecture given by Dr. Yamauchi at the 2019 EGU Assembly in April. As such, it has an undeniable legitimacy and gravitas and should be published. I learned a lot from reviewing it.

This reviewer is unfamiliar with the details of the Medal citation or the charge to the speaker for the presentation. However, the identified focus seems narrow to me. It begins by identifying as a focus "findings by Cluster that are not covered by review papers on circulations [itemized reviews]."

The list of reviews is missing a few significant ones that should be included, e.g. Lotko, 2007, Moore and Horwitz, 2007, Maggiolo 2016.Most if not all of the identified reviews are at least a decade old, so it would seem more appropriate to focus the paper as a review of this important topic in space science, with significant recent developments from all relevant missions, as well as relevant theory papers.

[Figure]

The manuscript has some significant problems ranging from cosmetic and language to substantive, as described below. These should be addressed in a revision.

The top level issue concerns the referencing in this paper. It cites 22 papers with Yamauchi as first author and an additional 12 with him as co-author, out of about 125 -130 papers cited. Some of the self-authored papers do not appear fully relevant to the present paper. For example: Yamauchi et al. 1996b, 1996c, 2006a, 2006b, 2009b, 2018a, Ohtani et al., 1995, Lundin et al. 1995, Ebihara et al., 2001. A balance should be struck in self-referencing, making sure all are strictly relevant to the present paper.

Meanwhile the manuscript fails to cite a number of relevant papers on the following topics: * Outflow flux correlation with solar wind parameters, e.g. Pollock et al., 1990 JGR, Moore et al., 1999 GRL. * Ion trajectory simulations of the the magnetosphere in idealized fields, e.g. Delcourt et al. 1990 JGR and a number of more recent papers. * Ion trajectory simulations with ionospheric ions in MHD fields, e.g. Fok et al. 2010 JGR, Moore et al., 2009 JGR, and related others. * Multifluid global circulation models with outflow in MHD fields, e.g. Glocer et al., 2009 JGR, Garcia-Sage et al., 2015 JGR, Brambles, et al. 2011 JGR, Wiltberger 2010 JGR.

With attention to those and the following issues, this paper will indeed serve the space science community well as a comprehensive review of relevant recent work on the title topic.

COSMETIC ISSUES: Technical editing for English usage is recommended for this paper, but here are some examples of the many issues.

L9: "in the other *hemisphere*?

L12: "unanswered questions"?

L16: "than that entering the magnetotail"?

L25: specie => species

L39: "wide circulation, ranging from the tail. . ."?

L46: Sue => Su

L210: "a mixture of different ions"?

L227: "to deconvolve different sources. . ."?

L300: sever => "severe"

L536: "ionospheric cusp, which is the destination for the entire mass-loading area."?

L601: convolute => "mix together"

L613: "have wide aspects" =>"extend broadly", physic =>physics

L622: "needed to explore the questions raised by these observations"?

L633: un-understood => poorly understood

CONTENT ISSUES:

Introduction section:

The rationale given for a focus on ions to the exclusion of electrons strikes this reviewer as questionable. Electrons and ions are inseparable because of quasi neutrality and the ambipolar electric field, such that electron heating also leads directly to ion escape [Strangeway et al., 2005]. Acknowledging that would not change much about the paper, but it seems so important that it should be at least discussed.

Outline:

An outline is given at the outset, which begins well, but then singles out the inner magnetosphere for selective treatment. It seems that this should be treated as one of the several significant "destinations of the outflow", rather than occupying its own major section. For example, total loss from the system may be a more important destination.

Sec.2 Outflow: The statement that cold filling flows have "never been directly detected"

is incorrect. See for example, Singh and Horwitz [1992 JGR], Watanabe et al., [1992 JGR], and others.

This statement: "...but most of them have high O+ outflow flux with much higher velocity than this cold supersonic outflow, and they are actually the suprathermal described below, including the apogee observation by Su et al. (1998)" is also incorrect. The polar wind at high altitude [Moore et al., 1997 Science] is inescapably mixed by the velocity filtering effect with auroral zone outflows, but these are readily discriminated by their higher energy and temperature, as summarized in the cited review by Moore et al. 1999.

The space devoted to the Freja (Figure 2) and Viking (Figure 14) observations of suprathermal and hot outflows seems inconsistent with the stated focus on "findings by Cluster...not otherwise reviewed [itemized reviews]." These could be condensed out of the paper by citing appropriate papers, since it is neither recent nor from Cluster. With a current figure count of 17, and many data figures, the paper is perhaps a bit overloaded with graphics at present.

The claim in lines 88-91 that "...the value is underestimated (by Su et al.) because the upper energy threshold of the instrument was 350 eV" is incorrect. Simultaneous observations by TIMAS at higher energies established that the core distribution was within the energy range for the events analyzed. The complementarity of these two instruments is well illustrated in Cladis et al. 1999 JGR.

Fig.3 is only cited far out of sequence in the text of line 554, p.28. Discussion of Fig. 3 should note that it agrees fairly well with earlier findings from DE1 (Pollock et al. 1990 JGR), Polar (Moore et al. 1999 GRL), and perhaps Geotail (Nose et al., 2005). It's not clear if these papers are covered by the reviews cited, and so should be cited in relation to this figure and paper by Schilling.

Destinations of outflow:

3.4 Secondary destinations?

Figure 5 (mislabeled Fig. 6 in Table 1) should better highlight the distinct difference between outflows from the auroral zones (day and night) and from the polar cap at higher latitudes, and the plasma trough region at lower latitudes, together with the velocity filtering effect that spreads the slower outflows relative to the faster auroral outflows. Also, the plasmasphere label "detach" should perhaps be on the dayside where the Plasmaspheric plume forms through the circulation effect, and detached blobs often are created [with references to Grebowsky 1970, Chappell 1972 and the IMAGE mission, Sandel et al.]

4 Inner magnetosphere:

Figures 7,8,9 get very short descriptions in line 220, and are simply pointed to in cited papers, so do not seem necessary in this review. Or perhaps a single example would do in place of three.

Figure 11 is quite striking and important, showing how low energy and high energy populations mix and become combined, while being dispersed by the velocity filtering effect.

4.4 Direct injection: This topic remains somewhat controversial, and the discussion should present both sides of the argument, with citations. The statement is made that these ions having the field aligned appearance of an outflow need not be inside the source/loss cone because of pitch angle diffusion. However, to demonstrate that this is the case requires observations of equal or higher fluxes inside the source/loss cone on the same flux tubes. The reviewer is not aware of such observations, but if they exist, they should certainly be cited here.

4.7 Loss Process: An effort is made here to explain how loss to the magnetosheath can occur "without reconnection", seemingly suggesting that reconnection would not work to explain such leakage, despite the existence of a huge recent body of observations

from Polar [Chandler et al., 1999 JGR, 2003 JGR, 2008 GRL] and MMS [Burch et al. 2016, Torbert et al., 2018, many others] showing that reconnection is a common occurrence, on the dayside magnetopause, leading to mixing of magnetosheath and cold ionospheric plasmas. In fact, an entire subcategory of MMS research has investigated in detail the ingestion and acceleration of cold terrestrial ions into dayside reconnection diffusion regions [Toledo-Redondo 2016 GRL]. This reviewer suggests that a critique of reconnection at the dayside magnetopause is beyond the scope of this manuscript and should not be tackled, implied or even suggested.

5 Consequences:

5.2 O+ escape effects on solar wind interaction: here the suggestion is made that the presence of O+ escaping from the magnetosphere can have a significant dynamic effect on the system as it is picked up to solar wind motion, by mass loading it down, and can thereby affect the energetics of the interaction as suggested in cited papers.

The discussion after Figure 16 cites the reconnection mechanisms of Dungey and Axford and Hines. However, Axford and Hines explored mainly a "viscous" interaction and ignored reconnection if my memory is correct. Suggest deleting that reference in the context of reconnection.

Considering the discussion of mass flux and reconnection rate, reference should be made here to the MMS papers on this topic, e.g. the most recent, [Fuselier et al. 2019 JGR] and perhaps some key references therein. The Vasyliunas [1995] concept of two nested dayside boundaries is new to me, but it doesn't seem applicable to a layered separation of magnetosheath, magnetospheric, and cold ionospheric plasmas. On the other hand, an inner boundary separating solar from terrestrial plasmas has been termed the "geopause" by Moore and Delcourt 1995 Reviews Geophysics. Also it has been used to organize global simulation work by Winglee 1998, 2002, and more recently by Wiltberger et al. 2015, Liemohn et al. 2016, and Glocer et al., 2018. It should be useful in the present paper when discussing inner and outer plasma boundaries.

The arguments leading to the conclusion of a need for a dedicated space mission to study these phenomena are well-posed and important and I applaud them.

---

## Referee Comment (RC2) · Anonymous Referee #2 · 5 Aug 2019

This paper is interesting in many aspects, it puts into perspective the importance of terrestrial ion outflow and dynamics in various parts of the earth's magnetosphere and examines the consequences of O+ escape into space, now and on geological scale. This paper deserves a quick publication in Annales Geophysicae after some revisions as indicated below:

Geotail, Cluster, Themis and later MMS have revealed Âń cold Âż ionospheric ions hidden by the spacecraft potential almost everywhere into the outer magnetosphere. This should be more clearly emphasized. For example, Seki used periods when Geotail was into eclipse and therefore was losing its positive charge to detect low energy ions in the anti-solar tail. This work is quoted in the paper, however Hirahara (2004), using the same spacecraft, evidenced the periodic emergence of multi composition cold ions modulated by geomagnetic field line oscillations in the dawn, near-Earth (10 Re), magnetosphere (Hirahara et al., JGR, 2004) and this work is not quoted. Similarly, Sauvaud

[Figure]

(2001) using Cluster showed the appearance, close to the dayside magnetopause, of a cold ion population of ionospheric origin when the magnetopause is put into motion by pressure changes (Sauvaud et al., Annales Geophysicae, 2001). The ionospheric ions can be very abundant, with densities up to 100 cm$-3$ and these cold ions mass load the magnetosphere, changing global parameters of magnetic reconnection, like the Alfvén speed or the reconnection rate. In addition, they introduce a new length-scale related to their gyroradius and kinetic effects which must be accounted for (André and Cully, GRL, 2012; Toledo-Redondo et al., GRL, 2016). These findings exemplified the importance of independent density measurement devices, i.e. relaxation sounder, electric field, waves. . .for spacecraft orbiting in the outer magnetosphere.

About the energy dependence of ion drift in the inner magnetosphere and related effects, several observations and interpretations have been made from missions before Cluster. For example, the ion gap resulting from the drift and residence time allowing charge exchange with neutrals close to the mirror points have been studied by Kovrazhkin et al. (Annales Geophysicae, 1998), and two types of ion gaps were reported by Buzulukova et al. (Annales Geophysicae, 2002).

The dynamics of hot ions injected from the tail plasma sheet is presented with a bias toward the inner magnetosphere. There are numerous studies concerning sporadic injections of hot magnetospheric H+ and O+ ions into the auroral bulge with accompanying strong ejections of ionospheric ions from the poleward border of the active auroral oval. Some of these works performed with Akebono, Interball and Cluster should be quoted (e.g., Hirahara, JGR, 1997, Sauvaud et al., JGR, 1999, Sergeev et al., JGR, 2000, etc.).

Finally, the author states correctly that the shadowing effect of the magnetopause and charge exchanges with the atmospheric neutrals close to the mirror points are two main loss mechanisms in the inner magnetosphere. Some words should be added on the losses occurring from plasmoid ejection during substorms which lead to the detection of escaping molecular and atomic ions in the far tail, as seen onboard Geotail and

STEREO (Christon et al., GRL, 1998; Kistler et al., JGR, 2010; Opitz et al., JGR, 2014, etc.)

Altogether the suggested corrections should not modify the structure of the paper.

---

## Author Comment (AC1) · 14 Aug 2019

Thank you very much for valuable comments with references. I try to implement individual comments as much as possible (e.g., dropping Figure 16 and 4) within the 25 page limit for this proceeding-type paper of the Medal lecture.

Before revising the paper (and before answering individual comments), I need to define the general direction toward the revision. My motivation for both lecture and this paper is condensed §5 and §4:

(1) Give good estimate for the ion loss rate from viewpoints of terrestrial evolution (§5.1), to show that it might have substantially affected on the evolution or fluctuation of the biosphere (including to stop the paradigm of "geomagnetic field protect the Earth from escape", §5.3) and ionosphere plays active role even at the high-altitude cusp due to the mass loading effect (e.g., test-particle method may mislead the cusp dynamics,

[Figure]

§5.2). The last viewpoints is the only realistic argument in proposing future missions to ESA because the magnetospheric/ionospheric "physics" cloud not be used for the argument for future missions under Cosmic Vision, and most likely in Voyage 2050. Relevant proposals and white paper are found at

http://www.irf.se/%7Eyamau/future/m5/ESCAPE_M5_Proposal_submitted.pdf
http://www.irf.se/%7Eyamau/future/f1/Fate_phase1_final.pdf
http://www.irf.se/%7Eyamau/future/whitepaper_i_n_final.pdf

(2) Showing importance of low-energy ion circulation in the inner magnetosphere, for which much less scientists worked during past 2 decades compared to the crowded fields such as magnetotail, radiation belt, and magnetopause (this is the main reason that self-siting references becomes too much and I will reduce them, §4). Since the return rate (lifetime of ions) strongly depends on energy, local energizations in the inner magnetosphere are included as much as possible if they are never introduced in review papers.

To stress on these points with 25 page limit, I have to skip many new development on the ion circulation such as what reviewers suggested (e.g., simulation, tail dynamics). Therefore, I would like to keep the present structure (and I noticed that title does not really much with the content). In this sense, I am flexible in changing the title such as "Ion loss process from the Earth" or something more adequate.

The make sure my motivation, I would revise the beginning of the Introduction as follows.

Introduction: Circulation and roles of ionospheric heavy ions have long been an important subject in the magnetospheric physics since they are found at almost everywhere in the magnetosphere (Chappell et al., 1972, Shelley et al., 1972) and even in the high-latitude magnetosheath and plasma mantle (Lundin, 1985, Eklund et al., 1997). There are many works on this subject with many reviews, but with relatively well covered subjects and less covered subjects. The former includes suprathermal ion outflow from

the ionosphere (Moore et al., 1999; Moore and Horwitz, 2007; Lotko, 2007; Darrouzet et al., 2009; Maggiolo, 2015; Welling et al., 2015), plasmaspheric cold ions (Darrouzet et al., 2009; Welling et al., 2015), plasma sheet dynamics including energization in the magnetotail and resultant ring current ions in the inner magnetosphere (Blanc et al., 1999; Walker et al., 1999; Ebihara and Ejiri, 2003; Kronberg et al., 2014), fate of cold supersonic ion outflow (AndreÌĄ, 2015) and the inner magnetospheric low-energy hot ions that are directly supplied from the auroral ionosphere (Moore et al., 1999). However, very little reviews discussed the total amount of the ionospheric ions to the space and its effect on the evolution of the planetary atmosphere (Moore and Horwitz, 2007; AndreÌĄ, 2015), which is a large contrast from the cases for Mars (Jakosky et al., 2015). In this viewpoint, there are two subjects that have nearly no review during past two decades: the outflowing ions directly accessing the solar wind in the magnetosheath, cusp, and outer part of plasma mantle, and fate of trapped hot ions that have much lower energy than the ordinary ring current. The Cluster's orbit and instrumentation allowed us to investigate these missing subjects with statistical significance. This paper reviews these missing parts in the context of the total amount of ion escape to the space based on Bartels Medal lecture at EGU General Assembly 2019, with stress on finding by Cluster and are not covered by above reviews (direct escape from the high-latitude magnetosphere and fate of hot ions in the inner magnetosphere).

Since Cluster Ion Spectrometory (CIS) COmposition DIstribution Function (CODIF) is not designed to separate more than four main species H+, He++, He+, and atomic ions of the CNO grounp (ReÌĂme et al., 2001), all heavy ions are pedagogically called oxygen ions O+ as is in the conventional manner, although heavy ions include nitrogen ions N+, and molecular nitrogen ions N+2 as the secondary component that becomes significant during geomagnetic storms (Chappell et al., 1982; Craven et al., 1985; Hamilton et al., 1988; Yau and Whalen, 1992). The paper is organized as following

If my revision strategy with new introduction (with possible new title) is acceptable, I start to define how to implement individual comments my within the discussion period.

---

## Author Comment (AC2) · 16 Aug 2019

**Reply to Reviewer #2's comments**

Thank you again for valuable comments and the recommendation. While general direction of the revision (in response to Reviewer #1) was already posted separately, I here describe specific responses to your individual comments.

The direct detections of cold ions, other than in eclipse, will mentioned at the beginning of §3 (Hirahara et al., 2004; Sauvaud et al., 2001), in §3.2 (Hirahara et al., 2004), and §4.2 (Sauvaud et al., 2001), and §4.7 (Sauvaud et al., 2001; Toledo-Redondo et al., 2016). However, the related arguments on the mass-loading and scale length are difficult to add in the present context because §5.2 is dedicated for heavy ions but not H+ or He++ (majority of cold ions). Instead of Andre and Cully (2012), I would use Andre (2015) that has a similar content with a review nature.

[Figure]

For the ion gap observations, I would add these references in §5.1 (Kovrazhkin et al., 1999) and §5.3 (Buzulukova et al., 2002).

A sentence mentioning the return flow toward the auroral ionosphere (Hirahara et al., 1997, Sauvaud et al., 1999) will be added in §3.4.

Finally, the loss through the plasmoid (Christon et al., 1998; Kistler et al., 2010) as well as through continuous flow (Hirahara et al., 1996; Opitz et al., 2014) will be mentioned in §3.2. There contributions are already included in the estimate in §4.1.

---

## Author Comment (AC3) · 23 Aug 2019

**Reply to Reviewer 1's comment**

————————————————

Thank you again for valuable comments to improve the manuscript. While my intention of general direction of the revision is already described in an independent comment file, I hereby outline how to improve based on individual comments.

\* English: Institute's native speaker promised to check the manuscript after I myself will double-check it.

\* Roles of electrons: After the scope of this paper is explained in the extended introduction (as described in the separate answer), I will add a statement that this paper does not mention the acceleration mechanisms or simulations, like: "Acceleration mechanisms for the outflowing ions such as wave particle interaction, electrostatic field, centrifugal acceleration, are not covered in this paper and hence roles of electrons are not discussed. Similarly, numerical simulations are not covered."

* Outline expected from Title: I will expand the introduction to justify the outline (as described in the separate answer). I also consider making the title more specific, like: "Terrestrial ion escape and relevant circulation in space" or "Ion loss routes from the Earth"

* Cold refilling: Thank you for pointing out my failure. The field-aligned velocity of order of 1 km/s observed by these references will be mentioned in §2.1. I will also move the estimated total outflow (refill) flux from §5.1 to §2.1.

* Polar wind and suprathermal are well discriminated: Thank you for pointing out my failure. I will replace the paragraph, like: "The source of these supersonic cold ions are most likely the polar wind because the velocity is similar to the prediction of the polar wind models and observations (Pollock et al., 1990; Yau and Whallen, 1992; Moore et al., 1999a). These observations show substantial O+ content with slower bulk velocity compared to H+. The difference from the Cluster observations (no O+) can be explained by the velocity filter effect, by which slow O+ are convected toward lower latitude than fast H+ (Chappel et al., 1987). Also slow O+ stay at low-altitude, where the wave-particle interaction is the most active, much longer than fast H+. This makes O+ energized to suprathermal energy range (next subsection) and detectable by standard hot ion instruments. "

* Figures 2 (Freja) and 14 (Viking): I will drop Figure 14 (and even Figure 4 of Prognoz-7 if needed). In the original manuscript, I included them because they are published in subscription journals which are not open access. For Figure 2, I prefer to keep it if space is available because this is still the only event when both outflow and inflow (assuming north-south symmetry) are detected, showing mass-independent energization without the help of statistics, and because the accumulated knowledge from Cluster

made the presented interpretation (as one of 5 possibility in 2005 paper) solid. If I need more space, I will remove this figure. I will add a statement that Cluster contributed the interpretation.

* line 88-91 (Simultaneous observations by TIMAS on Su et al.): Thank you very much for additional information for higher energy on Figure 14 of Su et al. (1998). I will remove this energy threshold argument.

* Figure 3: I will cite these papers explicitly (Schilling paper shows the solar wind dependence found at low-altitude (citations) are valid even for escaping ions to the space in the plasma mantle).

* Figure 5 (summary drawing): I will sub-divide auroral region and dayside, while I keep the same structure, because (1) I selected the most important route in estimating the total loss rate and (2) the suggested classification by the source has been done in many papers while only few works have been done with a classification by destinations that is important for total loss estimate.

* Figures 7-9: All figure are used in sections 4.3, 4.4, and 4.5. Figure 8 and Figure 9 (and could even Figure 7) will be moved to these sections.

* §4.4 Direct injection: By using Figure 7 here (and mentioning the lay-tracing in the original paper), I will mention that Cluster has shown a case of outflowing O+ without any bounce has finite PA outside the loss cone. Figure 9 also show the direct injection from above the auroral bulge with wide pitch angles at the equatorial plane.

* §4.7 Loss process: I will add the role of the reconnection, like: "This leaking process does not require local magnetopause reconnection (Sibeck et al., 1987; 1999; Sauvaud et al., 2001; Marcucci et al., 2004), while it is most effective during strong sunward convection sustained by nightside activities or during magnetopause erosion, both of which take place during increasing solar wind dynamic pressure and southward turning of IMF, and hence, very often coexists with the local magnetopause reconnection (Toledo-Redondo et al., 2016)"

* §5.2(a) O+ escape effect on solar wind interaction: I will remove the Axford reference (viscous like interaction), and add the geopause, like: "The concept of such second openness, i.e., two-time definition of the open-closed boundary, in two-time definition of the open-closed boundary,the open-closed boundary, first by the solar wind access point to the magnetosphere, and second by the access point of the outflowing iono-spheric ions to the solar wind, is similar to multiple‐branch discontinuity model by Vasyliunas (1995) and geopause concept by Moore et al. (1995) because prosent model can be achieved by replacing the upward propagation of electrodynamic information of the ionosphere by ion motions. "

* §5.2(b) The effect on the reconnection rate: I will add Fuselier et al. (2017) at just before §5.1, like: "Along this context, dayside part of the consideration of the O+ outflow has also been improved by treating the cusp outflow separately (Glocer et al., 2018), or by considering mass loading at the low-latitude dayside magnetopause (Fuselier et al., 2017)."

Since the proposed process is independent from reconnection, the word of "reconnection rate" was used just for comparison of efficiency, and since the nearby reconnection from this mass loading region is anti-parallel reconnection in which the zero magnetic field cannot stop the solar wind inflow whereas Fuselier's works concerns component reconnection which is driven by compression (more close to nightside reconnection), I am afraid referring Fuselier (2019) may cause a misleading. I simply refer Sonnerup (1974) for the reconnection rate.

* §5.2(c) Simulation including O+: Thank you many for suggested papers. This section deals only the direct feedback from the mass loading in the cusp-mantle region, while existing multi-fluid models considered feedback trough magnetotail (e.g., Wiltberger et a., 2015, 2017; Liemohl et al., 2016, 2018; Welling and Liemohl, 2016). Even Glocer et al. (2018) with good consideration of the cusp outflow did not treat such local

feedback. Therefore, I will mention only Winglee's work (pioneer on this matter) and Glocer's work.

---

## Author Response (AR1)

* * *
**Reply to Reviewer #1's comment**

Thank you very much for valuable comments with important references to improve the manuscript. I made major revision based on the comments. I try to implement individual comments as much as possible (e.g., dropping Figure 16 and 4) while keeping the paper compact and follow the EGU lecture that is much based on my own works.

*> This manuscript is reported to be based on the Bartels Medal Lecture given by Dr. Yamauchi at the 2019 EGU Assembly in April. As such, it has an undeniable legitimacy and gravitas and should be published. I learned a lot from reviewing it.*
*> This reviewer is unfamiliar with the details of the Medal citation or the charge to the speaker for the presentation. However, the identified focus seems narrow to me. It begins by identifying as a focus "findings by Cluster that are not covered by review papers on circulations [itemized reviews]."*

To make the paper compact, I minimized the description on subjects that has been covered in past reviews, and tried to complement the past reviews by focusing on the subjects that have been overlooked, because studies on these overlooked subjects are one of the main reasons for the Medal. They are (1) entangling the zoo of hot-ion (50 eV - 5 keV) in the inner magnetosphere, (2) roles and pathways of ion escape in the total atmospheric escape from a planet and its importance on the atmospheric evolution even including the Earth, (3) other hidden importance of planetary ions that are found in the dayside magnetospheric boundary for Mars and the Earth. Therefore, I would like to keep the present structure.

To stress on these points in compact way, I skipped many new development on the ion circulation such as what reviewers suggested (e.g., simulation, tail dynamics). To make sure my motivation clear, I completely revised the beginning of the Introduction. Accordingly, I changed the title to "Terrestrial ion escape and relevant circulation in space".

*> The list of reviews is missing a few significant ones that should be included, e.g. Lotko, 2007, Moore and Horwitz, 2007, Maggiolo 2016. Most if not all of the identified reviews are at least a decade old, so it would seem more appropriate to focus the paper as a review of this important topic in space science, with significant recent developments from all relevant missions, as well as relevant theory papers.*

I added the suggested references in the introduction, and further classified then in terms of subjects. However, as mentioned above, I put stress on the subject what is the most poorly studied field, for which the Medal was given, and I try to make a "unique" review that is not a simple upgrade of past reviews. Simulations of outflow and magnetotail, and observations of MMS/Van-Allen probe results should be reviewed by experts on these subjects because, unlike the subject I cover, there are many scientists working on.

*> The manuscript has some significant problems ranging from cosmetic and language to substantive, as described below. These should be addressed in a revision.*

Below I describe how I have revised the manuscript.

*> The top level issue concerns the referencing in this paper.*
*>It cites 22 papers with Yamauchi as first author and an additional 12 with him as co-author, out of about 125-130 papers cited. Some of the self-authored papers do not appear fully relevant to the present paper. For example: Yamauchi et al. 1996b, 1996c, 2006a, 2006b, 2009b, 2018a, Ohtani et al., 1995, Lundin et al. 1995, Ebihara et al., 2001. A balance should be struck in self-referencing, making sure all are strictly relevant to the present paper.*

I removed some of old references that have less relevance to the present topic, while made the relevance clearer for what I decided to keep. With increased references of others, new title and new introduction, "poorly studied subject", I believed that balance is now ok.

*> Meanwhile the manuscript fails to cite a number of relevant papers on the following topics:*

I added total 31 new references.

> * Outflow flux correlation with solar wind parameters, e.g. Pollock et al., 1990 JGR, Moore et al., 1999 GRL.

I added a paragraph describing the correlation with the solar wind parameters in both §2.2 and §2.3 with comparison with the past relevant results.

> * Ion trajectory simulations of the magnetosphere in idealized fields, e.g. Delcourt et al. 1990 JGR and a number of more recent papers.
> * Ion trajectory simulations with ionospheric ions in MHD fields, e.g. Fok et al. 2010 JGR, Moore et al., 2009 JGR, and related others.
> * Multifluid global circulation models with outflow in MHD fields, e.g. Glocer et al., 2009 JGR, Garcia-Sage et al., 2015 JGR, Brambles, et al. 2011 JGR, Wiltberger 2010 JGR.

As I explained above (answer to the first comment), I intend to skip ion trajectory simulations from this paper to make the paper compact and unique.

> With attention to those and the following issues, this paper will indeed serve the space science community well as a comprehensive review of relevant recent work on the title topic.

Thank oyu very much

> COSMETIC ISSUES: Technical editing for English usage is recommended for this paper, but here are some examples of the many issues.
x >  L9: "in the other *hemisphere*?
x > L12: "unanswered questions"?
x > L16: "than that entering the magnetotail"?
x > L25: specie => species
x > L39: "wide circulation, ranging from the tail. . ."?
x > L46: Sue => Su
x > L210: "a mixture of different ions"?
x > L227: "to deconvolve different sources. . ."?
x > L300: sever => "severe"
x > L536: "ionospheric cusp, which is the destination for the entire mass-loading area."?
x > L601: convolute => "mix together"
x > L613: "have wide aspects" =>"extend broadly", physic =>physics
x > L622: "needed to explore the questions raised by these observations"?
x > L633: un-understood => poorly understood

For English, Institute's native speaker checked the manuscript after I myself double-checked it, including the above examples.

> CONTENT ISSUES:

**\* Introduction / Roles of electrons:**
> The rationale given for a focus on ions to the exclusion of electrons strikes this reviewer as questionable. Electrons and ions are inseparable because of quasi neutrality and the ambipolar electric field, such that electron heating also leads directly to ion escape [Strangeway et al., 2005]. Acknowledging that would not change much about the paper, but it seems so important that it should be at least discussed.

After the scope of this paper is explained in the extended introduction, I added a statement that this paper does not mention the acceleration mechanisms (including ambipolar diffusion) or simulations.

**\* Introduction / Outline:**
> An outline is given at the outset, which begins well, but then singles out the inner magnetosphere for selective treatment. It seems that this should be treated as one of the several significant "destinations of the outflow", rather than occupying its own major section. For example, total loss from the system may be a more important destination.

The introduction was extended to justify the outline and the specific stresses of this paper (§4 and §5). I also changed the title more specific, as "Terrestrial ion **escape and relevant** circulation in space
"

**\* Sec.2 Outflow / Cold refilling:**
*> The statement that cold filling flows have "never been directly detected" is incorrect. See for example, Singh and Horwitz [1992 JGR], Watanabe et al., [1992 JGR], and others.*

Thank you for pointing out my mistake. The field-aligned velocity of order of 1 km/s observed by these references is now mentioned in §2.1. I also moved the estimated total outflow (refill) flux from §5.1 to §2.1.

**\* Polar wind and suprathermal outflow:**
*> This statement: "...but most of them have high O+ outflow flux with much higher velocity than this cold supersonic outflow, and they are actually the suprathermal described below, including the apogee observation by Su et al. (1998)" is also incorrect. The polar wind at high altitude [Moore et al., 1997 Science] is inescapably mixed by the velocity filtering effect with auroral zone outflows, but these are readily discriminated by their higher energy and temperature, as summarized in the cited review by Moore et al. 1999.*

Thank you for pointing out my mistake. I revised the entire paragraph and mentioned the velocity filter effect with references added.

**\* Figures 2 (Freja) and 14 (Viking):**
*> The space devoted to the Freja (Figure 2) and Viking (Figure 14) observations of suprathermal and hot outflows seems inconsistent with the stated focus on "findings by Cluster...not otherwise reviewed [itemized reviews]." These could be condensed out of the paper by citing appropriate papers, since it is neither recent nor from Cluster. With a current figure count of 17, and many data figures, the paper is perhaps a bit overloaded with graphics at present.*

I dropped Figure 14. I also dropped Figure 4 (Prognoz-7). For Figure 2, I would keep it because this is still the only event when both outflow and inflow (assuming north-south symmetry) are detected on the same traversal, showing mass-independent acceleration, and because the accumulated knowledge from Cluster made the presented interpretation (as one of 5 possibility in 2005 paper) solid. However, if my paper exceed the limit what the editor allows me, this is the first figure to remove.

For the Cluster relevance, I tried to add a statement like "This scenario was just one of several scenarios in Yamauchi et al. (2005b), and is now the most likely one after Cluster observations", but removed again because this destroys the flow of the paper.

**\* TIMAS observation by Su et al:**
*> The claim in lines 88-91 that ". . .the value is underestimated (by Su et al.) because the upper energy threshold of the instrument was 350 eV" is incorrect. Simultaneous observations by TIMAS at higher energies established that the core distribution was within the energy range for the events analyzed. The complementarity of these two instruments is well illustrated in Cladis et al. 1999 JGR.*

Thank you for additional information for higher energy on Figure 14 of Su et al. (1998). I removed the energy threshold argument, and revised a large part of text by referring Akebono observation.

**\* Figure 3:**
*> Fig.3 is only cited far out of sequence in the text of line 554, p.28. Discussion of Fig. 3 should note that it agrees fairly well with earlier findings from DE1 (Pollock et al. 1990 JGR), Polar (Moore et al. 1999 GRL), and perhaps Geotail (Nose et al., 2005). It's not clear if these papers are covered by the reviews cited, and so should be cited in relation to this figure and paper by Schilling.*

I moved the figure to §3.3 and added paragraphs to introduce the solar wind (+ F10.7 and Kp) dependence and comparison with low-altitude observations with more relevant and recent citations for dayside outflow (Moore et al., 1999a; Cully et al., 2003; Lennartsson et al., 2004).

**\* Destinations of outflow:**

*> Figure 5 (mislabeled Fig. 6 in Table 1) should better highlight the distinct difference between outflows from the auroral zones (day and night) and from the polar cap at higher latitudes, and the plasma trough region at lower latitudes, together with the velocity filtering effect that spreads the slower outflows relative to the faster auroral outflows.  Also, the plasmasphere label "detach" should perhaps be on the dayside where the Plasmaspheric plume forms through the circulation effect, and detached blobs often are created [with references to Grebowsky 1970, Chappell 1972 and the IMAGE mission, Sandel et al.]*

I sub-divided the auroral region and dayside in §3.3. There are now labeled as (b2), and old label (b) becomes (b1).  However, I kept the same structure (dividing by energy rather than place), because (1) the suggested classification by the source has been done in many papers while only few works have been done with a classification by destinations that is important for total loss estimate, and (2) the most important route in estimating the total loss rate is dayside, and the ambiguity mainly comes from incomplete estimate of energization.  Accordingly, the description for the nightside outflow is kept minimum.  For the amount, I added a paragraph in §3.3.

**\* Figures of inner magnetosphere:**
*>4 Inner magnetosphere:*
*> Figures 7,8,9 get very short descriptions in line 220, and are simply pointed to in cited papers, so do not seem necessary in this review. Or perhaps a single example would do in place of three.*

All three figure are used in §4.3, §4.4, and §4.5, and are moved to there.  Instead, I referred "example" figure (Figure 5).

*> Figure 11 is quite striking and important, showing how low energy and high-energy populations mix and become combined, while being dispersed by the velocity filtering effect.*

I moved this figure (now Fig. 8) up and enhanced the explanation, while I moved the other simulation (dynamic one, now Fig. 9) right after this figure to enhance the usefulness of Cluster-simulation comparison.

**\* 4.4 Direct injection:**
*> This topic remains somewhat controversial, and the discussion should present both sides of the argument, with citations. The statement is made that these ions having the field-aligned appearance of an outflow need not be inside the source/loss cone because of pitch angle diffusion. However, to demonstrate that this is the case requires observations of equal or higher fluxes inside the source/loss cone on the same flux tubes. The reviewer is not aware of such observations, but if they exist, they should certainly be cited here.*

I am not aware of such high-pitch angle observations because the loss cone in the equatorial plane is less than 1 degree.  I asked Arase team to install such capability but they could not.  However, by using Fig. 10h-10g (and mentioning the lay-tracing in the original paper), I mentioned that Cluster has shown a case of outflowing O+ without any bounce although they are outside the loss cone.  Fig. 11 also shows the direct injection from above the auroral bulge with wide pitch angles at the equatorial plane.

**\* 4.7 Loss process:**
*> An effort is made here to explain how loss to the magnetosheath can occur "without reconnection", seemingly suggesting that reconnection would not work to explain such leakage, despite the existence of a huge recent body of observations from Polar [Chandler et al., 1999 JGR, 2003 JGR, 2008 GRL] and MMS [Burch et al. 2016, Torbert et al., 2018, many others] showing that reconnection is a common occurrence, on the dayside magnetopause, leading to mixing of magnetosheath and cold ionospheric plasmas. In fact, an entire subcategory of MMS research has investigated in detail the ingestion and acceleration of cold terrestrial ions into dayside reconnection diffusion regions [Toledo-Redondo 2016 GRL]. This reviewer suggests that a critique of reconnection at the dayside magnetopause is beyond the scope of this manuscript and should not be tackled, implied or even suggested.*

I added short description of the role of the reconnection: it is not required for leaking but it often coexists with the local magnetopause reconnection.

**\* 5 Consequences**

> *5.2 O+ escape effect on solar wind interaction: here the suggestion is made that the presence of O+ escaping from the magnetosphere can have a significant dynamic effect on the system as it is picked up to solar wind motion, by mass loading it down, and can thereby affect the energetics of the interaction as suggested in cited papers.*

Thank you for briefing.

**\* Axford and Hines paper:**
> *The discussion after Figure 16 cites the reconnection mechanisms of Dungey and Axford and Hines. However, Axford and Hines explored mainly a "viscous" interaction and ignored reconnection if my memory is correct. Suggest deleting that reference in the context of reconnection.*

I removed the Axford reference (viscous like interaction).  I meant Axford and Hines as "remote-loading by the ionosphere" because energy consumption in the ionosphere of "moving polar magnetosphere" cannot distinguish between the global reconnection and viscous-like interaction.  But such a detail is not needed to present in this paper.

**\* The effect on the reconnection rate:**
> *Considering the discussion of mass flux and reconnection rate, reference should be made here to the MMS papers on this topic, e.g. the most recent, [Fuselier et al. 2019 JGR] and perhaps some key references therein.*

I added Fuselier et al. (2017) at just before §5.1.  Since the proposed process is independent from reconnection, the word of "reconnection rate" was used just for comparison of efficiency, and since the nearby reconnection from this mass loading region is anti-parallel reconnection in which the zero magnetic field cannot stop the solar wind inflow whereas Fuselier's works concerns component reconnection which is driven by compression (more close to nightside reconnection), I am afraid referring Fuselier (2019) may cause a misleading.  I simply referred Sonnerup (1974) for the reconnection rate.

**\* Vasyliunas paper:**
> *The Vasyliunas [1995] concept of two nested dayside boundaries is new to me, but it doesn't seem applicable to a layered separation of magnetosheath, magnetospheric, and cold ionospheric plasmas.*

I rephrased by adding the geopause concept there.  With both aspects referred, readers may understand the similarity and differences between the mass-loading boundary and the previous multi-layer/multi-wave concepts by Moore et al. (1995) and Vasyliunas (1995).

**\* (d) Simulation including O+:**
> *On the other hand, an inner boundary separating solar from terrestrial plasmas has been termed the "geopause" by Moore and Delcourt 1995 Reviews Geophysics. Also it has been used to organize global simulation work by Winglee 1998, 2002, and more recently by Wiltberger et al. 2015, Liemohn et al. 2016, and Glocer et al., 2018. It should be useful in the present paper when discussing inner and outer plasma boundaries.*

Thank you for suggested papers.  This section deals only the direct feedback from the mass loading in the cusp-mantle region, while existing multi-fluid models considered feedback trough magnetotail (e.g., Wiltberger et a., 2015, 2017; Liemohl et al., 2016, 2018; Welling and Liemohl, 2016).  Even Glocer et al. (2018) with good consideration of the cusp outflow did not treat such local feedback.  Therefore, I mentioned only Winglee's work (pioneer on this matter) and Glocer's work.
* * *
**Reply to Reviewer #2's comments.**

Thank you very much for valuable comments and recommendation with important references to improve the manuscript.

> *This paper is interesting in many aspects, it puts into perspective the importance of terrestrial ion outflow and dynamics in various parts of the earth's magnetosphere and examines the consequences of O+ escape into space, now and on geological scale. This paper deserves a quick publication in Annales Geophysicae after some revisions as indicated below:*

Below I describe how I have revised the manuscript based on Reviewer#2's comment. In addition, I made a major revision based on reviewer #1's comment.

> *Geotail, Cluster, Themis and later MMS have revealed "cold" ionospheric ions hidden by the spacecraft potential almost everywhere into the outer magnetosphere. This should be more clearly emphasized. For example, Seki used periods when Geotail was into eclipse and therefore was losing its positive charge to detect low energy ions in the anti-solar tail. This work is quoted in the paper, however Hirahara (2004), using the same spacecraft, evidenced the periodic emergence of multi composition cold ions modulated by geomagnetic field line oscillations in the dawn, near-Earth (10 Re), magnetosphere (Hirahara et al., JGR, 2004) and this work is not quoted. Similarly, Sauvaud (2001) using Cluster showed the appearance, close to the dayside magnetopause, of a cold ion population of ionospheric origin when the magnetopause is put into motion by pressure changes (Sauvaud et al., Annales Geophysicae, 2001). The ionospheric ions can be very abundant, with densities up to 100 cm,àí3 and these cold ions mass load the magnetosphere, changing global parameters of magnetic reconnection, like the Alfvén speed or the reconnection rate. In addition, they introduce a new length-scale related to their gyroradius and kinetic effects which must be accounted for (André and Cully, GRL, 2012; Toledo-Redondo et al., GRL, 2016). These findings exemplified the importance of independent density measurement devices, i.e. relaxation sounder, electric field, waves. . .for spacecraft orbiting in the outer magnetosphere.*

The direct detections of cold ions, other than in eclipse, is now mentioned at the beginning of §3 (Hirahara et al., 2004; Sauvaud et al., 2001), in §3.2 (Hirahara et al., 2004), in §4.2 (Sauvaud et al., 2001), and in §4.7 (Sauvaud et al., 2001; Toledo-Redondo et al., 2016).

The related arguments on the mass-loading and scale length are difficult to add in the present context because §5.2 is dedicated for heavy ions but not H+ or He++ (majority of cold ions).

Instead of Andre and Cully (2012), I used Andre (2015) that has a similar content with a review nature.

> *About the energy dependence of ion drift in the inner magnetosphere and related effects, several observations and interpretations have been made from missions before Cluster. For example, the ion gap resulting from the drift and residence time allowing charge exchange with neutrals close to the mirror points have been studied by Kovrazhkin et al. (Annales Geophysicae, 1998), and two types of ion gaps were reported by Buzulukova et al. (Annales Geophysicae, 2002).*

For the ion gap observations, I added these references in §5.1 (Kovrazhkin et al., 1999) and §5.3 (Buzulukova et al., 2002).

> *The dynamics of hot ions injected from the tail plasma sheet is presented with a bias toward the inner magnetosphere. There are numerous studies concerning sporadic injections of hot magnetospheric H+ and O+ ions into the auroral bulge with accompanying strong ejections of ionospheric ions from the poleward border of the active auroral oval. Some of these works performed with Akebono, Interball and Cluster should be quoted (e.g., Hirahara, JGR, 1997, Sauvaud et al., JGR, 1999, Sergeev et al., JGR, 2000, etc.).*

The direct return flow toward the auroral ionosphere (Hirahara et al., 1997, Sauvaud et al., 1999) was added in §3.4.

> *Finally, the author states correctly that the shadowing effect of the magnetopause and charge exchanges with the atmospheric neutrals close to the mirror points are two main loss mechanisms in the inner magnetosphere. Some words*

*should be added on the losses occurring from plasmoid ejection during substorms which lead to the detection of escaping molecular and atomic ions in the far tail, as seen onboard Geotail and STEREO (Christon et al., GRL, 1998; Kistler et al., JGR, 2010; Opitz et al., JGR, 2014, etc.)*

Finally, the losses through the plasmoid (Christon et al., 1998; Kistler et al., 2010) and through the continuous flow (Hirahara et al., 1996; Opitz et al., 2014) are mentioned in §3.2. Their contributions are already included in the estimate in §4.1.